# Injection into and extraction from single fungal cells

Orane Guillaume-Gentil [1✉], Christoph G. Gäbelein[1], Stefanie Schmieder [1,3], Vincent Martinez[2], Tomaso Zambelli[2], Markus Künzler [1] & Julia A. Vorholt [1✉]

The direct delivery of molecules and the sampling of endogenous compounds into and from living cells provide powerful means to modulate and study cellular functions. Intracellular injection and extraction remain challenging for fungal cells that possess a cell wall. The most common methods for intracellular delivery into fungi rely on the initial degradation of the cell wall to generate protoplasts, a step that represents a major bottleneck in terms of time, efficiency, standardization, and cell viability. Here, we show that fluidic force microscopy enables the injection of solutions and cytoplasmic fluid extraction into and out of individual fungal cells, including unicellular model yeasts and multicellular filamentous fungi. The approach is strain- and cargo-independent and opens new opportunities for manipulating and analyzing fungi. We also perturb individual hyphal compartments within intact mycelial networks to study the cellular response at the single cell level.

[1] Institute of Microbiology, ETH Zurich, 8093 Zurich, Switzerland. [2] Institute for Biomedical Engineering, ETH Zurich, 8092 Zurich, Switzerland. [3] Present address: Division of Gastroenterology, Boston Children's Hospital, Boston, MA 02115, USA. ✉email: gorane@ethz.ch; jvorholt@ethz.ch

Fungi play pivotal ecological and economical roles in nature, agriculture, medicine, and for industrial applications, from the degradation of dead organic material to their beneficial role in plant growth and the production of many drugs and food products[1–5]. In addition, numerous plant and animal pathogenic fungi are known, many of which are difficult to combat[6]. They are also important model organisms in biological research to study the molecular biology of eukaryotes. The manipulation of fungi is thus crucial to deepen our understanding of fungal biology in general and to optimize their use in agriculture and bioproduction. However, in practice, the development of such manipulation techniques faces many obstacles. These are mainly due their complex cell wall structure, acting as a physical barrier, and to the enormous diversity of fungal species and thus the difficulty in developing universally applicable protocols.

A crucial aspect in fungal cell biology is the ability to deliver exogenous materials into living cells e.g., of DNA for transformation. Other cargos of interest include RNA, proteins, peptides, and small (anti) metabolites, as well as synthetic nanomaterials such as quantum dots or nanoparticles[7–9]. While effective delivery methods are available for animal cells, introducing exogenous materials into fungal cells has proven challenging, especially for filamentous fungi[7]. Bulk transformation approaches include protoplast-mediated or *Agrobacterium tumefaciens*-mediated transformation, electroporation, biolistics, and shock waves. However, most of these approaches show relatively poor efficiency[1]. Although fungal protoplasts are most widely used for transformation, cell wall removal requires dedicated protocols for each fungal species because they differ in cell wall composition and structure, and remains a challenge for most fungi. In addition, the formation of protoplast leads to a dramatic decrease in cell viability[1].

A promising alternative for efficient intracellular delivery relies on the use of nanopipettes to directly access the interior of an individual cell[10]. As a general concept, a fluidic nanoprobe is inserted into a cell to create a channel across the cell membrane, and molecules loaded in the probe are released into the cell. The approach also allows molecules to be captured from the cell into the nanoprobe. Thus, these technologies not only provide a straightforward universal approach for intracellular delivery, but also enable the sampling of endogenous molecules, offering a powerful means of analysing animal cells[11]. Fluidic force microscopy (FluidFM) has shown a great potential in this area. The FluidFM technology is a force-sensitive nanopipette that enables pressure-driven fluid handling in the femto- to pico-litre range[12,13]. We have previously demonstrated FluidFM-based injection of exogenous molecules into mammalian cells,[14] as well as the extraction of cytoplasmic or nuclear biopsies from living cells for molecular analyses[15,16].

However, the use of nanopipettes with fungi has shown only limited success so far, mostly hindered by the high rigidity of the cell wall, the difficulty of immobilizing the cell to prevent its lateral displacement during manipulation, as well as their relatively small cell size[17]. Riveline et al. trapped *Schizosaccharomyces pombe* in microstructures, and shear-damaged their cell wall locally while releasing solution with the same pipette, which resulted in some of the solution reaching the cell interior[18].

Here, we introduce a quantitative FluidFM-based method for injection into and extraction from fungi, including filamentous fungi and unicellular yeasts, that is widely applicable (Fig. 1a).

## Results

**Technological developments**. Fungi are characterized by a large diversity of life forms. In order to establish a broadly applicable platform for their manipulation, we selected four model organisms that cover their main distinctive morphological features: the unicellular fungi *Saccharomyces cerevisiae* and *Schizosaccharomyces pombe*, which are the two major models for budding and fission yeasts, respectively; the dimorphic yeast *Candida albicans*, an opportunistic pathogen that can switch from a unicellular budding growth to a multicellular mycelial growth under specific environmental conditions; and *Coprinopsis cinerea*, a multicellular filamentous fungus and model organism for studying fungal development and the evolution of multicellular fungi. *S. cerevisiae*, *S. pombe* and *C. albicans* belong to the phylum Ascomycota whereas *C. cinerea* is a representative of the phylum Basidiomycota of the kingdom Fungi[19].

The general workflow for injecting or extracting fluid into and from living cells using FluidFM consists of the following steps: (1) positioning the pyramidal FluidFM tip above a selected cell; (2) driving the tip through the cell wall and membrane using force spectroscopy; (3) maintaining the tip in the cell at a constant force while applying over- or underpressure for injection or extraction, respectively; (4) withdrawing the tip from the cell (Supplementary Fig. 1a). This approach works reliably with mammalian cells[14]. However, it is not readily applicable to fungal cells due to their unique physical properties, particularly the rigid cell wall.

The insertion of the FluidFM tip into a cell requires that the cell is immobilized on a hard substrate. While most animal cells spread and adhere to solid substrates, the adhesion of fungal cells is comparatively weak[20–22]. Therefore, we first evaluated whether fungal cells maintained a fixed position upon indentation as a prerequisite for injection. On flat substrates, the spherical yeast *S. cerevisiae* and the rod-shaped yeast *S. pombe* were laterally displaced upon contact with the FluidFM tip. We therefore designed microstructured substrates and could show that the yeast cells were spatially constrained (Supplementary Figs. 2, 3, "Methods"). In contrast, the hyphae of the dimorphic yeast *C. albicans* and of the filamentous fungi *C. cinerea* adhered sufficiently on glass and plastic polystyrene, respectively, to prevent cell displacement upon tip insertion.

Furthermore, the cell wall surrounding fungal cells results in a drastically increased cell rigidity, causing an increase in the forces required to insert a probe into the cells compared to animal cells (Supplementary Fig. 1b). The high stiffnesses of fungal cells were reflected in the force-distance curves obtained by indentation and eventual insertion of the probe inside fungal and animal cells (Fig. 1b). For effective probe insertion into a typical adherent mammalian cell (e.g., HeLa), the maximal force was set such that the tip crossed the entire cell to the underlying substrate, which typically required forces up to 200 nN[14]. In comparison, the force-distance curves on fungal cells showed that, while the indentation distances were similar or shorter than for HeLa cells, the forces required for maximal indentation were 10 times larger, i.e., up to 2 µN. FluidFM probes were thus re-designed, optimizing their length and thickness to increase their stiffness to ~2.3 ± 0.5 N/m, so that forces up to several micro-Newtons could be exerted.

Finally, cell size also represents a crucial aspect for the development of an injection protocol for fungal cells. It determines the upper injectable volumes, especially due to the cell wall, which provides a physical barrier to cell expansion. We estimated the cell volumes of the four selected fungi, in comparison to human HeLa cells (Fig. 1c). The mean (SD) cell volumes of the yeasts *C. albicans*, *S. pombe* and *S. cerevisiae* were 75 (50), 105 (25) and 80 (55) fL, respectively. The average volume of hyphal compartments of the mushroom *C. cinerea* was 360 (290) fL. In comparison, HeLa cells had a mean volume of 2500 (950) fL, approximately ten- to thirtyfold larger. Because the volumes injected into animal cells were typically on the order of

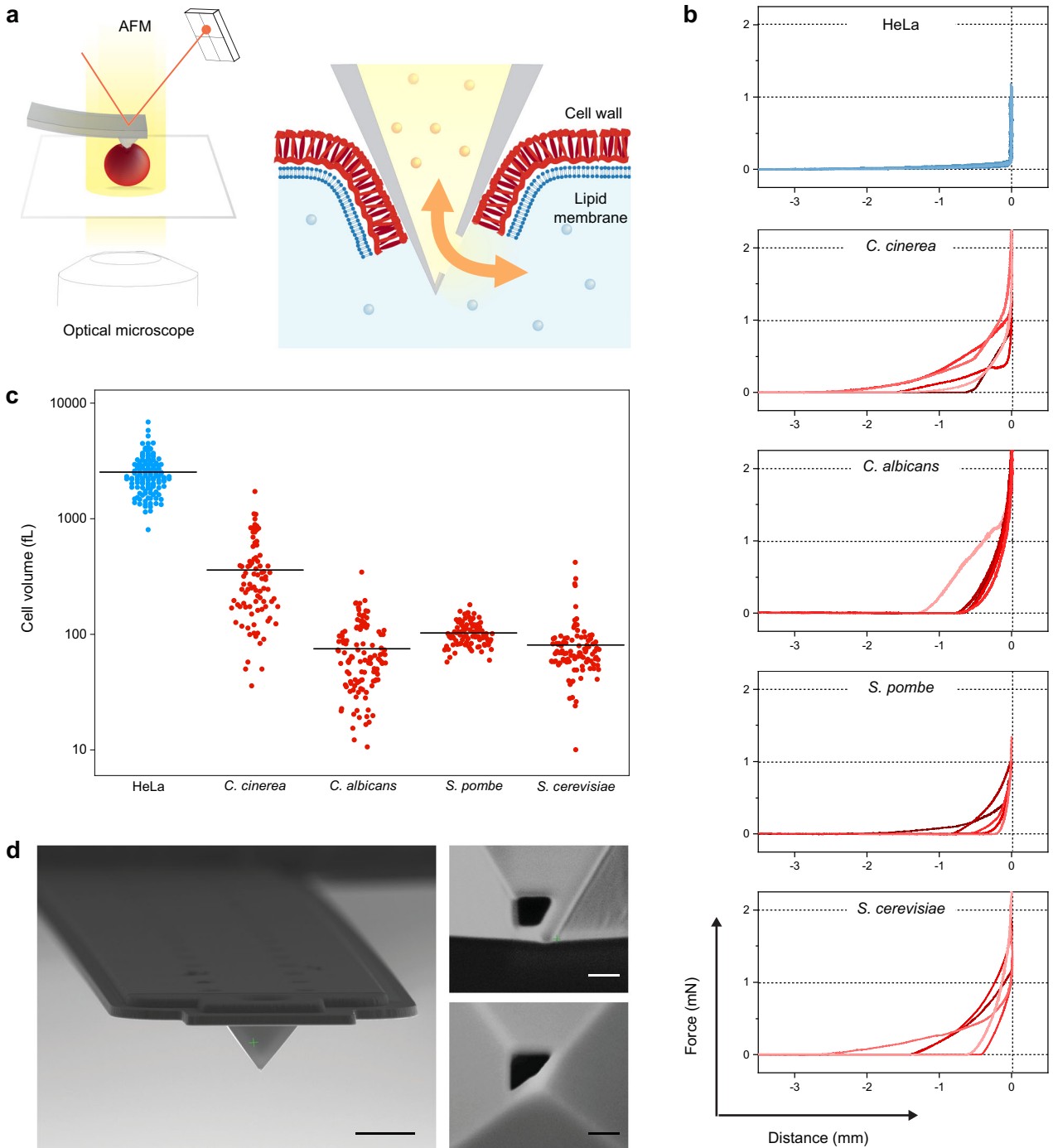

**Fig. 1 Technological developments. a** Scheme of FluidFM-based injection/extraction of fungi. A microfluidic probe operated with an atomic force microscope (AFM) is inserted through the hard cell wall, allowing for a pressure-driven exchange of liquids between the probe microchannel and the cell interior. The FluidFM is mounted on an optical microscope that allows in situ monitoring of the manipulation. **b** Representative force-distance curves obtained upon probe insertion in the different organisms. Each plot features 5 curves recorded with different cells. **c** Estimated cell volumes, as measured from 2D microscopy images assuming tubular and spherical cell geometries, for (dissociated) HeLa cells ($N = 132$), *C. cinerea* ($N = 88$), *C. albicans* hyphal compartments ($N = 116$), *S. pombe* ($N = 100$) and *S. cerevisiae* ($N = 105$). Horizontal lines are means. **d** Scanning electron microscopy images of FluidFM probes with a custom-designed tip aperture for fungal injection. Left: Front view of the hollow cantilever with a pyramidal tip. Scale bar: 5 μm. Right: side- (top) and bottom- (down) views of the aperture on the pyramidal tip. Scale bars: 200 nm.

hundreds of femtoliters[14], we anticipated that volumes injectable into fungal cells will be on the order of tens of femtoliters. Cell size is also relevant in the context of proper probe insertion, as the probe aperture must be sufficiently narrow and close to the pyramid apex to be fully within the cell just upon puncturing. Yet, at the same time, the aperture area should be large enough for

sufficient flow. We thus shaped probe apertures with focused ion beam embracing two of the four sides of the tip, with aperture sizes down to 200 nm in height and width (Fig. 1d).

**Fluorescent tracer injection**. To assess the ability to cross the cell wall and inject solutions into fungal cells and to adjust the

injection parameters, we first used the membrane-impermeable fluorescent tracer Lucifer yellow (LY). We performed force spectroscopy on the fungal cells and subsequently monitored LY fluorescence in real time during and after applying a pressure difference of $+1000$ mbar for several minutes. Using a fluorescent probe allowed detection of both cell staining resulting from a successful injection and extracellular leakage in the event of a compromised tip insertion (Supplementary Fig. 4). Consistent with the force-distance curves (Fig. 1b), the application of a force of $2.0 \pm 0.1\,\mu N$ resulted in 100% successful injection of the dye into fungal cells without apparent leakage ($N = 9$), indicating full insertion of the probe aperture into the cell (Fig. 2). Of note, when we tested the injection with lower forces, i.e., $1.0 \pm 0.1\,\mu N$ and $1.5 \pm 0.1\,\mu N$, this resulted in intracellular staining, but with concomitant leakage of the fluorophore into the extracellular environment in 100% of the cases ($N = 11$), and 83% ($N = 18$), respectively. The simultaneous release of LY both inside the cell and in the extracellular environment indicated effective cell wall pervasion but incomplete insertion of the probe aperture. With applied forces lower than $1.0 \pm 0.1\,\mu N$, the cell wall was not pierced, and the fluorophore was released entirely in the extracellular medium. These results highlight the need to apply sufficient force for robust injection into fungal cells, which we achieve with 2 μN and represents an important step from earlier studies[18].

Following injection, the unicellular yeasts *S. pombe* and *S. cerevisiae* were uniformly labeled, whereas neighboring yeasts remained unlabeled (Fig. 2a, b). Although homogenous staining may also result from extracellular binding of the fluorophore to the cell wall, the selective labeling of the targeted but not the neighboring yeasts indicated successful delivery of the fluorophore inside the cell.

Unequivocal intracellular targeting of the fluorophore was achieved by injection of hyphal and pseudohyphal cells of *C. albicans* (Fig. 2c). While single cells of the dimorphic yeast are similar in size to *S. pombe* and *S. cerevisiae* (see Fig. 1c), *C. albicans* forms multicellular structures under certain environmental conditions. Upon injection of hyphal germ tubes, a homogenous staining of the entire structure was observed, as expected. In fact, germ tubes are hyphal projections that develop in the first cell cycle, before septation. When injecting hyphal compartments, the released fluorophore selectively stained the targeted compartments and did not cross the mature septa. In addition, injection of pseudo-hyphae resulted in labeling of some but not all compartments, in agreement with the dye passing through primary but not mature septa. In addition, vacuoles in the injected compartments remained unlabeled, validating the cytoplasmic location of the injected fluorescent tracer.

Similarly, in *C. cinerea* mycelia, the injected fluorophore dispersed throughout the targeted hyphal compartments, included their branches, but did not pass through the septa delimiting the cells, indicating effective intracellular injection (Fig. 2d). The intracellular delivery of LY was further validated by injecting vacuolated hyphal compartments, whereby the injected dye clearly located in the cytoplasmic fluid and was excluded from the vacuoles (Fig. 2d, right panels).

Next, we estimated the volumes of solution that we injected based on the fluorescence intensities of the injected cells. The volumes were similar for the four organisms and ranged between 1 and 100 femtoliters (Supplementary Fig. 5). We note here that eventual extracellular staining of the cell wall may result in additional fluorescence and the injected volumes may thus be slightly overestimated. For comparison, volumes up to 900 fL were injected in HeLa cells[14]. The estimated injected volumes were thus consistent with the respective sizes of fungal and animal cells described above.

Regarding the pressure pulse applied to release the fluorophore solution in the cell, we applied a pressure difference up to more than 1000 mbar, during several minutes. In mammalian cells, such pressure pulses resulted in injection volumes larger than 1 pL, which led to inflation, membrane rupture and cell death. In the fungal cells, no cell deformation was observed, consistent with the rigid cell wall maintaining cell shape and the high internal pressure of the cells.

**Post-injection cell viability**. The results described above showed that injection into fungal cells is indeed feasible. The next set of experiments was aimed to assess whether the method preserves cell viability. We therefore monitored the growth of the fungi following injection. In addition, we used a fluorescently-labeled histone H1 (H1-AF488) injected into *S. cerevisiae* and *C. cinerea*. When the exogenous nuclear protein is delivered into the cell, it is expected to be actively transported to the nucleus in fully functional cells.

In the unicellular *S. cerevisiae*, the injected histone protein effectively accumulated in the nuclei, while a diffuse fluorescence signal was observed in the cytoplasm and cell wall; moreover, the fluorescent protein was excluded from vacuoles (Fig. 3a). After injection of a budding mother cell, the labeled histone was transported into the nucleus of the mother and the daughter cells, with a lower fluorescence intensity in the daughter cell (Fig. 3b). Monitoring of the injected yeasts by time-lapse microscopy showed that the injected cells were moving, and continued to produce a daughter cell in the hours following injection (Fig. 3c). After the budding event, the labeled histone was observed in both nuclei of the mother and daughter cells (Fig. 3d).

Similarly, in the filamentous fungi *C. cinerea*, the injected histone proteins were effectively transported into the nuclei, while slightly labeling the walls of the cells (Fig. 4a). The homokaryon strain of *C. cinerea* most frequently contains two nuclei per hyphal segment[23]. Consistent with this, we mostly observed the staining of two nuclei per injected hyphal compartments, with a weaker stain for the nucleus most distant from the injection site (Fig. 4a). Vegetative mycelia of *C. cinerea* grow by apical extension of hyphae and the formation of subapical branches, each of which becomes an apically elongating hypha. To evaluate post-injection growth, we injected hyphal tip cells and subsequently monitored them by time-lapse microscopy (Fig. 4b). Apical growth was observed for all injected hyphal tip cells, with a similar elongation rate as for non-injected hyphae. Upon mycelial growth, the injected H1-AF488 remained in its location and observable for at least 2 days.

The observation that the injected histone protein accumulated in nuclei not only validated the successful intracellular injections, in both uni- and multicellular fungi, but also demonstrated the short-term preservation of cellular functions. Further monitoring of fungal growth after injection confirmed the absence of adverse effect on cell viability.

**Extraction**. Next, we evaluated the ability to extract cellular fluid from fungal cells. We used an oil-prefilled FluidFM probe to allow confinement and visualization of cytoplasmic fluid extracted into the probe. Tip insertion into *C. cinerea* hyphal compartments was performed as for injection, but instead of applying overpressure, we applied underpressure for the suction of the cell content. When exerting underpressure, we effectively observed the collection of fluid in the cantilever probe (Fig. 5a, inserts). When targeting a hyphal tip cell, continued growth of the hypha was observed after tip insertion, but stopped after aspiration of the cytoplasmic fluid into the probe (Fig. 5a, blue arrows). The extraction was accompanied by the appearance of vacuoles in the

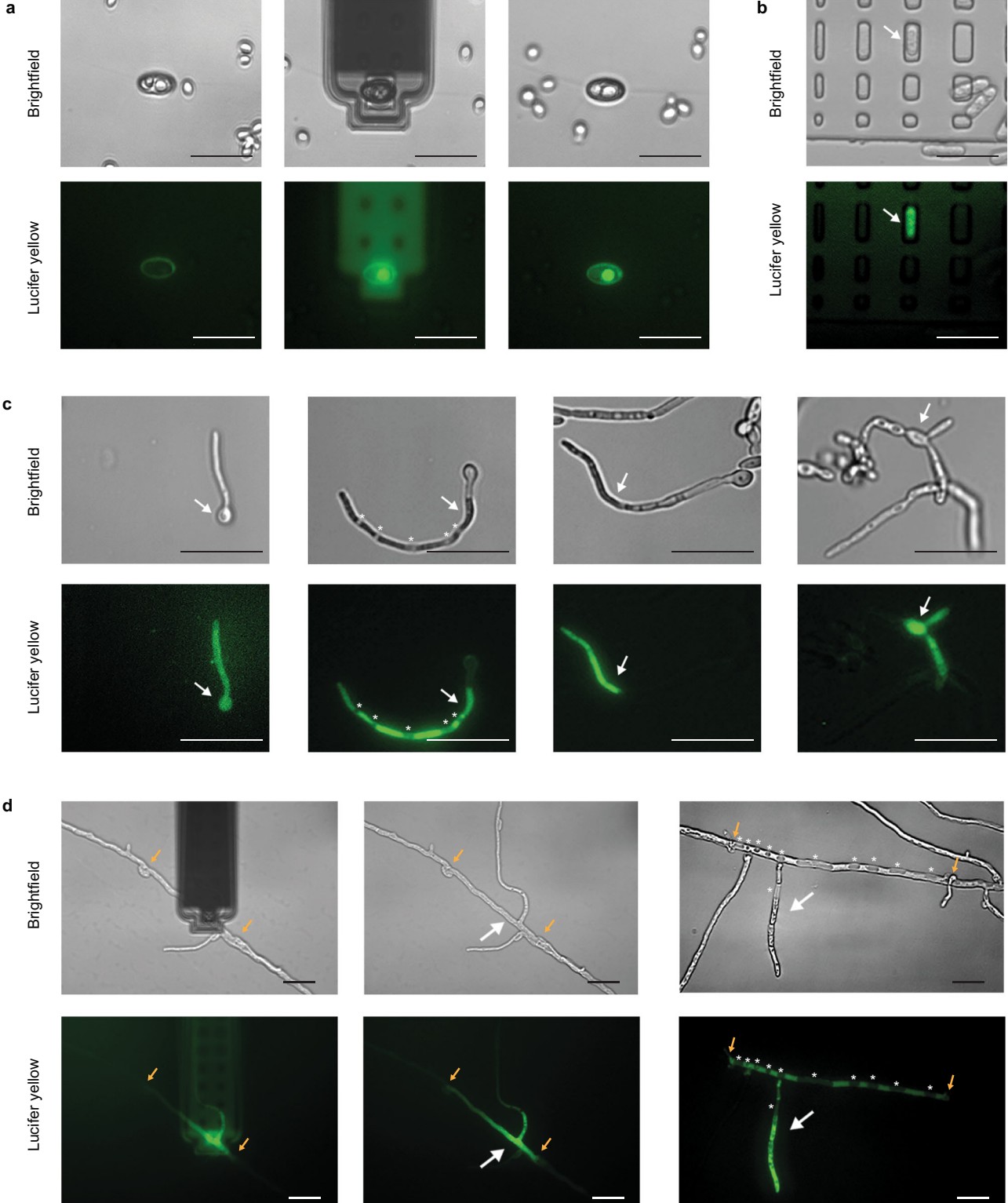

**Fig. 2 Fluorophore injection. a** Bright field and fluorescence images of the injection of lucifer yellow (LY) into *S. cerevisiae*, before, during, and after the injection. The images show a single oval microwell (in the center) containing two yeast cells; the right cell in the microwell was injected and was labeled, while the adjacent cell to its left was not injected and remained unlabeled. **b** *S. pombe* injected with LY. **c** Germ tube (left), hyphae (center left, center right), and pseudo-hyphae (right) of *C.albicans* injected with LY. **d** LY injections in *C. cinerea* hyphal compartments. The white arrows in (**b**), (**c**), and (**d**) indicate the injection sites. The asterisks in (**c**) and (**d**) indicate the vacuoles. The orange arrows in (**d**) indicate the septa of the injected hyphal compartment. Scale bars are 20 μm.

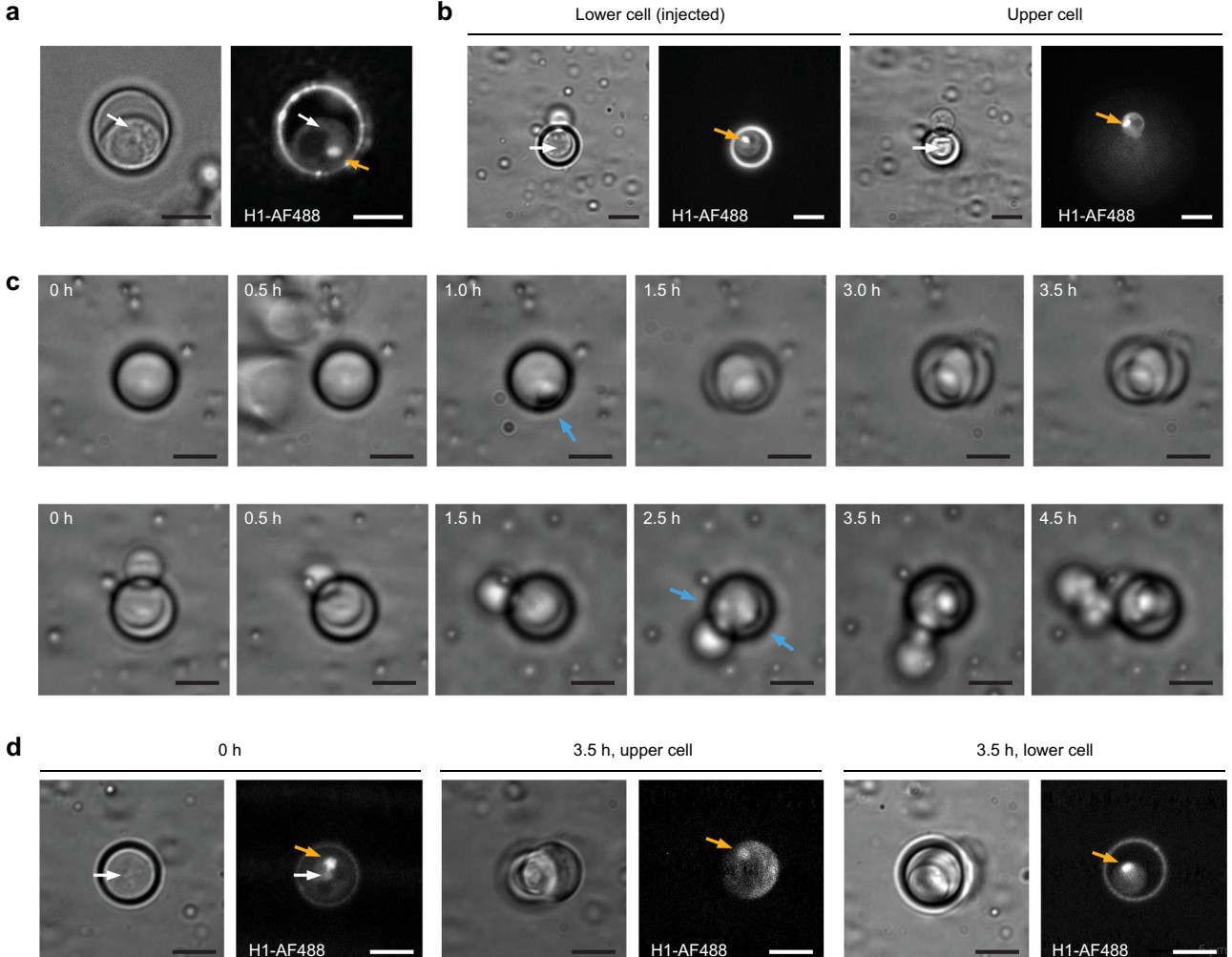

**Fig. 3 Post-injection viability of *S. cerevisiae*. a** H1-AF488 injected in *S. cerevisiae* accumulated in the cell nucleus (orange arrow), and was excluded from the vacuole (darker). The white arrow shows the injection site. **b** Following injection of a budding *S. cerevisiae* (lower cell), the labeled histone accumulated in both the lower and the upper cell nuclei (orange arrows). The white arrows show the injection site. **c** Time-lapse brightfield images of *S. cerevisiae* after injection. The first raw shows the growth post-injection of a single yeast. Budding was observed 1 h after the injection (blue arrow), and 2 cells were then observed on the following time frames. The second raw shows the growth of a yeast that was budding at the time of injection (same cell as in **b**). The injected yeast continued its growth, budding ~2.5 h after injection (blue arrows), and resulting in 4 cells at later time frames. **d** After division of an injected single cell, the labeled histone was found in the nuclei of both the mother and the daughter cells. The white arrows show the injection site, the orange arrows show the labeled nuclei. All the images were created by summing the slices of a Z-stack. Scale bar: 5 μm.

targeted hyphal compartment. No morphological changes were observed in the adjacent hyphal compartments (Fig. 5a). These observations indicated successful extraction of cytoplasmic fluid.

**Examining single cells within a multicellular fungus.** After demonstrating extraction from individual hyphal cells, we applied this method to address an open question in *C. cinerea* research. The filamentous fungus represents a prey for fungivorous nematodes, such as *Aphelenchus avenae*[24]. It has been shown that *A. avenae* feeds on *C. cinerea* by perforation of the cell wall with its stylet and suction of the cellular fluid[25]. This attack in turn triggers a defense response of *C. cinerea* that involves the expression of the defense gene *cgl2*[25,26]. Previous results suggest that the physical cue of penetration and suction is necessary to trigger *cgl2* expression but it remained unclear whether it is sufficient or whether chemical cues, such as the concomitant release of molecules by the fungivorous nematode, are needed[27]. We hypothesized that extraction with the FluidFM could mimic nematode feeding and thus, allow us to test whether

hyphal content removal is sufficient to induce the fungal defense response. To address this question, we used a reporter strain of *C. cinerea* carrying a dTomato expression cassette driven by the promoter of the *cgl2* gene[25]. Volumes ranging from 210 fL up to 1750 fL were extracted from different hyphal compartments, and the mycelia was then monitored overnight for the expression of the fluorescent reporter. Time-lapse fluorescence microscopy after extraction showed an increasing dTomato signal, selectively in the hyphal compartments that had been extracted (Fig. 5b). Quantitative measurements of the fluorescent reporter confirmed a steady increase of the reporter fluorescence over time within hyphal compartments that had been extracted, whereas neighboring cells, which did not undergo extraction, did not express the fluorescent reporter (Fig. 5c). This suggests that the suction of cytoplasm was sufficient to induce a defense response in *C. cinerea*. As pointed out above, we had observed that single hyphal cell extraction resulted in growth arrest. It is thus interesting to note that the cells remained viable and maintained the capacity to express the fluorescent reporter protein, despite the

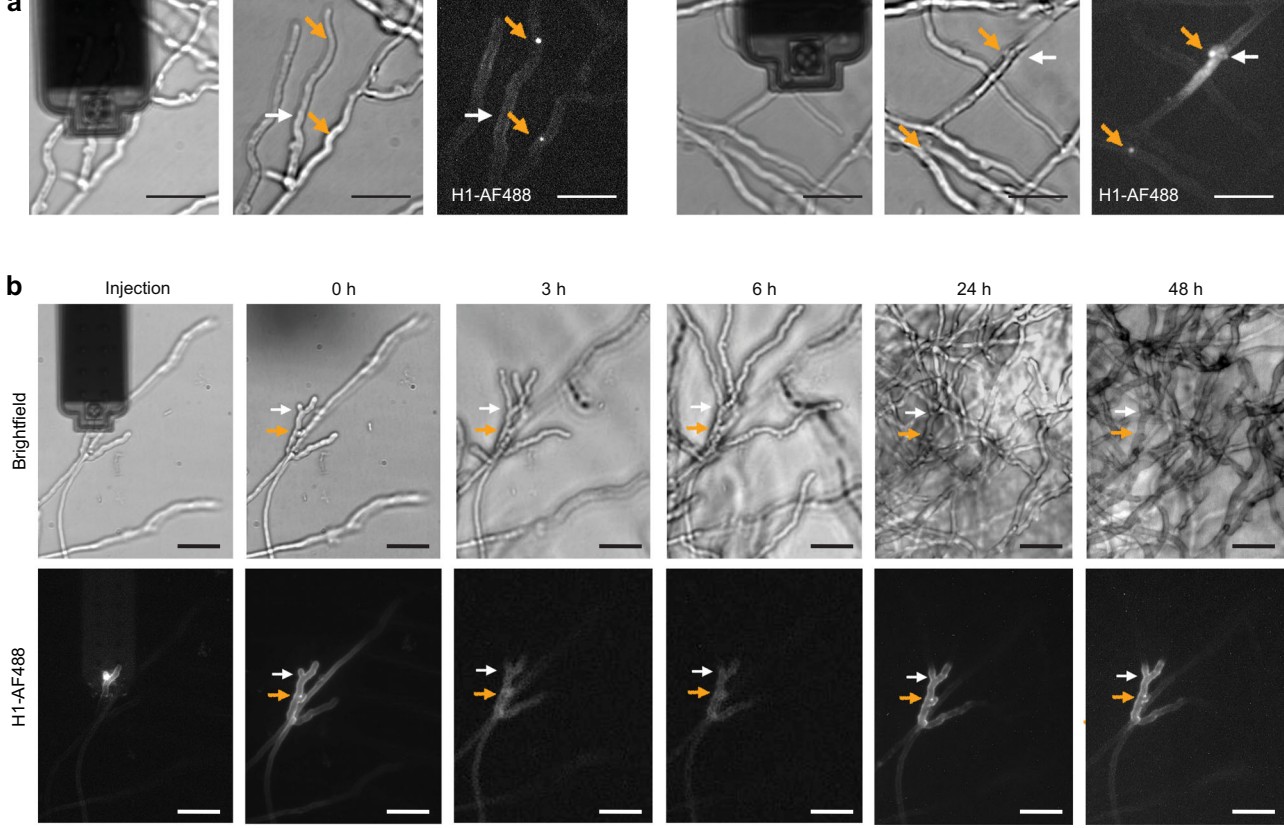

**Fig. 4 Post-injection viability of *C. cinerea*. a** H1-AF488 injected in *C. cinerea* accumulated in the cell nuclei (orange arrows). Two representative injections are shown. The white arrow shows the injection sites. **b** Following injection of a hyphal tip, time-lapse monitoring showed the growth of the injected cell, similar to the surrounding hyphae. The injected H1-AF488 remained at its location along the whole time-course. The fluorescence images in (**a**), and the brightfield and fluorescence images at 0, 24, and 48 h in (**b**) are summed slices of Z-stacks. Scale bar: 20 µm.

important loss of cytoplasmic fluid. A previous study revealed that upon localized challenge with nematodes, the fungal defense response was generally confined to the site of predation, but it propagated over several millimeters in a distinct subset of specialized hyphae, characterized by a large diameter[25]. While such "trunk" hyphae were not targeted in the present study, the FluidFM extraction approach will provide an attractive tool to investigate the propagation of the fungal defense response at the single-cell resolution.

## Discussion
The method developed in this work allows traversing the cell wall of living fungi with a nano-fluidic probe and either delivering a solution into individual fungal cells or extracting cytoplasmic fluid. It provides a straightforward universal technique for intracellular delivery that is adapted to virtually any natural or synthetic molecules that can be dispersed in solution as well as to the diversity of fungal species. We demonstrated proof-of-concept delivery of a fluorescent dye and a labeled protein into uni- and multi-cellular fungi, and further showed sustained cell viability and growth after the injection. Numerous applications are conceivable in which the injected cargo enables either monitoring or interfering a particular endogenous molecular process in living single cells or multicellular networks. A relatively large cell-to-cell variation in the injected volumes was observed, similar to what has been reported for injections into mammalian cells[14,28]. Therefore, determination of the injected volumes can only be done post-injection, by fluorescence measurements. If the molecule of interest does not have fluorescence, it can be injected together with a fluorescent marker. Unlike mammalian

cells, direct injection into fungal nuclei was not attempted as they were too small for insertion of the FluidFM tip. In addition to intracellular delivery, the method also enables extraction of cytoplasmic fluid from single fungal cells. In this work, we applied this technique to mimic nematode feeding and study the fungal defense response to this predation. This proof-of-concept study illustrates the ability to manipulate and analyze hyphal cells within intact mycelial networks at single-cell resolution, opening new avenues for studying cell–cell communication and organization in filamentous fungi. Furthermore, with emergent developments in bioanalytics, it will also be possible to use the approach to analyze the molecular content of individual fungal cells, similar to studies emerging for mammalian cells[11].

## Methods
**Cell line and culture**. Hela cells (ATCC) were maintained in high-glucose DMEM medium (Life Technologies, Carlsbad, CA) supplemented with 10% fetal bovine serum (FBS) (Gibco), and 1× penicillin/streptomycin solution (Life Technologies) in a 5% CO2 humidified atmosphere at 37 °C.

**Fungi strains and media**. Yeast extract-peptone-dextrose (YPD) medium was prepared with 1% (w/v) yeast extract (Oxoid), 2% (w/v) bacto peptone (BD) and 2% (w/v) glucose (Sigma). Yeast maltose and glucose (YMG) medium was prepared with 0.4% (w/v) yeast extract (Oxoid), 1.0% (w/v) malt extract, and 0.4% (w/v) glucose (Sigma). Defined synthetic dextrose (SD) medium was prepared with 0.19% (w/v) yeast nitrogen base without amino acids and ammonium sulfate (For-Medium), 0.082% (w/v) complete supplement mixture of amino acids (For-Medium), and 2% (w/v) glucose (Sigma). Phosphate-buffered saline (PBS) was prepared with 0.8% (w/v) NaCl (Merck), 0.02% (w/v) KCl (Sigma), 0.144% (w/v) Na2HPO4 (Sigma) and 0.024% (w/v) KH2PO4 (Sigma), adjusted to pH 7.4.

Wild-type *Schizosaccharomyces pombe* and *Saccharomyces cerevisiae* (W303) were grown in YPD medium. For injection experiments, 5 mL cultures were grown overnight at 180 rpm and a temperature of 28 °C. A 100 µL of the cell

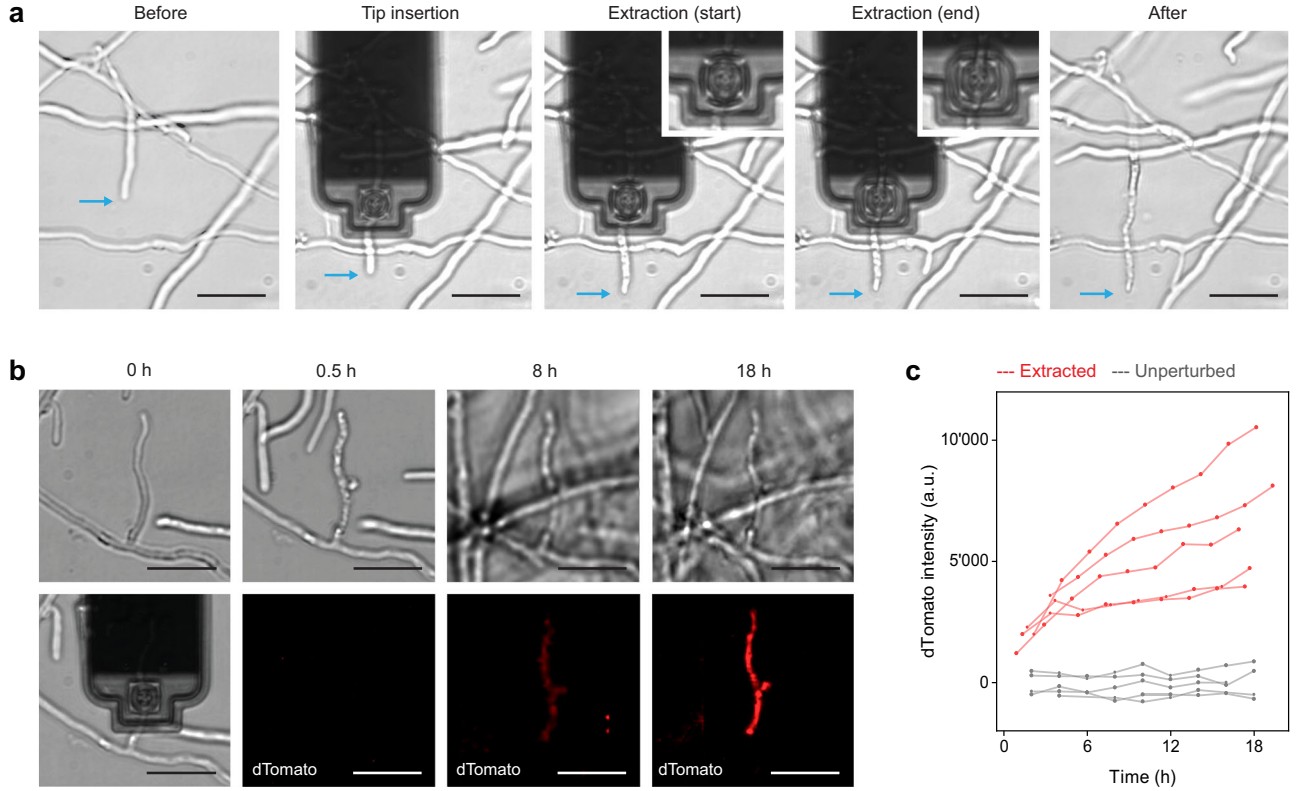

**Fig. 5 Cytoplasmic fluid extraction from *C. cinerea*. a** Brightfield images showing the extraction of cytoplasmic fluid from *C. cinerea*. Upon application of suction, the cellular fluid is collected in the front of the FluidFM probe (inserts). The targeted hyphae was growing until suction was applied for extraction (blue arrows). Vacuoles are visible after the extraction, exclusively within the targeted compartment. **b** Extraction (0 h) and subsequent monitoring of the dTomato fluorescent reporter. **c** Quantified dTomato fluorescence intensity of individual hyphal compartments over time after extraction. Red curves profiles were obtained from extracted hyphal compartments ($N = 5$). Gray curves are dTomato expression profile from control, non-perturbed hyphal compartments within the same mycelia ($N = 5$). Scale bars: 20 μm.

suspension was seeded onto the microstructured substrates. For initial experiments (fluorophore injection), 5 mL of PBS were added. For later experiments (cell viability), 5 mL of SD medium were added. The cells were let sediment onto the substrate for at least 15 min.

Wild-type *Candida albicans* (SC5314) were grown in YPD medium. For injection experiments, 5 mL cultures were grown overnight at 180 rpm and a temperature of 28 °C. To induce yeast-to-hyphal transition, 1 mL of the cell suspension was then resuspended in 4 mL PBS and supplemented with 1 mL foetal bovine serum (Chemie Brunschwig AG). The cells were then cultured for about 5 h at 37 °C and 180 rpm. 500 μL of the cell suspension was then seeded onto a 50 mm glass bottom dish (low side wall, WillCo Well B.V.) and 5 mL PBS were added.

Vegetative mycelia of *Coprinopsis cinerea* wildtype strain (A43mutB43mut; homodikaryon) and of the reporter strain A43mutB43mut *cgl2p*-dTom (expression of cytoplasmic dTomato under control of *cgl2p*) were cultivated on YMG agar plates at 37 °C in the dark. Samples for injection and extraction experiments were prepared as follow: a plug was inoculated onto the cover (~low side wall) of a 50 mm-petri dish. 2 mL of YMG liquid medium were added and the fungi were let grown at 37 °C in the dark for 4–5 days. Fresh medium was added every 1–2 days. Shortly prior to the experiment, 4 mL low-fluorescence SD medium was added to the dish.

**Reagents for injection**. HEPES-2 buffer was prepared with 4-(2-hydroxyethyl) piperazine-1-ethanesulfonic acid (HEPES, 10 mM), supplemented with sodium chloride (150 mM) in Millipore water, with a pH adjusted to 7.4 using NaOH (6 M). Lucifer Yellow CH dipotassium salt (LY, Sigma–Aldrich Chemie GmbH), and Alexa488-labeled histone H1 (Invitrogen) were used at concentrations ranging from 0.5 to 2.0 mg/mL in HEPES-2 buffer. All solutions were filtered through 0.2 μm pore size filters before use.

**Micro-structured substrates**. Atomic force microscopy studies (AFM) of microorganisms require their immobilization on the substrates, otherwise they roll away as soon as the AFM tip enters in contact with their cell wall[29,30]. Immobilization strategies were pioneered by the Dufrêne's group taking advantage of porous polymer membranes[31]. For this study, we designed substrates with microstructures of different lateral and height dimensions and assayed their efficacy for

the immobilization of unicellular yeasts upon indentation with a FluidFM probe. The micro-structured substrates were produced by Wunderlichips GmbH, and were made of Kraton A1536 polymer, with the microstructures laying on a $40 \pm 5$ μm thick bottom layer. Successful immobilization of *S. cerevisiae* was obtained with circular and oval micro-wells, with a depth of 5 μm, and lateral dimensions ranging from 5 to 12 μm. Successful immobilization of *S. pombe* was obtained using trenches or rectangle micro-wells, with a depth of 3 μm, a width of 4–5 μm, and a length of 12 μm or longer. 1 cm × 1 cm arrays of microstructures were laid on a drop of 100% ethanol onto a 50 mm glass bottom dish (Willco Well B.V.). The overall thickness of the sample allowed the use of objectives with high numerical apertures.

**Optical microscopy**. We used an inverted AxioObserver microscope equipped with a temperature-controlled incubation chamber (Zeiss), and coupled to a spinning disc confocal microscope (Visitron) with a Yokogawa CSU-W1 confocal unit and an EMCCD camera system (Andor). Phase-contrast and fluorescence images were acquired using 10×, 40× (0.6 na), and 63× (1.4 na, oil) objectives and a 2× lens switcher using VisiView software (Visitron).

Lucifer yellow CH and Alexa Fluor 488-Histone H1 were imaged using a 488 nm laser and a 525/50 emmision filter. The dTomato fluorescent reporter was imaged using a 561 nm laser and a 609/54 nm emission filter.

Microscopy images were analyzed using the AxioVision and Fiji softwares. When indicated (Figure caption), fluorescence images were created by summing the slices of a Z-stack (maximum intensity projection) in Fiji.

**FluidFM setup**. We used a FluidFM system composed of a FlexAFM-NIR scan head and a C3000 controller driven by the EasyScan2 software (Nanosurf), and a digital pressure controller unit (ranging from −800 to +1000 mbar) operated by a digital controller software (Cytosurge). A syringe pressure kit with a three-way valve (Cytosurge) was used in addition to the digital pressure controller to apply larger pressure differences.

FluidFM Rapid Prototyping probes made of silicon nitride were obtained from Cytosurge. The probes were coated with an 18 nm carbon layer using a CCU-010 Carbon Coater (Safematic), and the aperture was milled using a FIB-SEM Nvision 40 device (Zeiss) equipped with a gallium ion source using the Atlas software

(Zeiss). The front face of the pyramidal probe was aligned perpendicularly to the FIB-beam, and the aperture, extending on two faces of the pyramidal tip, was milled (see Fig. 1d) with an acceleration voltage of 30 kV at 10 pA (100 nm thick silicon nitride layer at the aperture region). The processed probes were then mounted onto a cytoclip holder by Cytosurge.

For experiments with *S. cerevisiae*, *S. pombe*, and *C. cinerea*, the FluidFM probes were plasma treated for 1 min (PDG-32G, Harrick Plasma) and coated with the hydrophobic, anti-fouling SL2 Sigmacote (Sigma–Aldrich) on the inside and outside with heat stabilization for ≥ 45 min at 100 °C prior to use (see[15] for a detailed protocol), before filling the probe with the solution to inject.

For experiment with *C. albicans*, the probes were plasma treated for 1 min (PDG-32G, Harrick Plasma), filled with the solution to inject, and then coated with a hydrophilic, anti-fouling polymer as follows: the probes were immersed for at least 45 min in a solution of poly(L-lysine)-graft-(polyethylene glycol) (PLL-*g*-PEG, with a PLL backbone chain of 20 kDa, 2 kDa PEG side chains and a grafting ratio of 3.5, obtained from SurfaceSolutionS GmbH) at 0.5 mg/mL in Millipore water. The probes were then immersed for at least 5 min in Millipore water for rinsing.

**Estimation of fungal cell volumes**. All cell dimensions were measured in the AxioVision software, using brightfield images acquired with a 40× objective. HeLa cells were dissociated by trypsinization before imaging.

The cell volumes were calculated from the measured cell diameters, assuming a spherical shape for HeLa cells and *S. cerevisiae*, and from the measured length and width, assuming a tubular shape for *S. pombe* and for the hyphal compartments of *C. albicans* and *C. cinerea*.

**Injection experiments**. Cantilever spring constant was measured using a software-implemented calibration module (Nanosurf), with the resonance frequencies and quality factors obtained from thermal noise spectrum acquired in air, before filling of the probe microchannel. The solution to inject was loaded in the probe reservoir, and the probe was then connected to the pressure controller unit to apply an overpressure $\Delta p$ of 1000 mbar to flow the solution from the reservoir into the microchannel. The cantilever sensitivity was then calibrated in the sample experimental medium (PBS or SD medium).

The cell to be injected was visualized by phase contrast microscopy, and the FluidFM probe was approached on the substrate in the vicinity of the targeted cell with a force set point of 20 nN and retracted with the Z-piezo. The probe was then laterally displaced above the desired point of insertion, under observation by phase contrast microscopy. The tip of the probe was then inserted into the cell through a forward force spectroscopy routine driven by the Z-piezo with a velocity of 1000 nm s$^{-1}$. The forward force spectroscopy was set to stop when reaching a maximal force (Fmax) of up to 2000 nN. The probe was then maintained in the cell interior at constant Fmax for 1 up to 30 min. During this pause, an overpressure pulse $\Delta p$ of 1000 mbar or higher was applied to deliver the solution. The probe was then retracted through a backward force spectroscopy with a velocity of 1000 nm s$^{-1}$. The entire injection process was monitored by phase contrast and fluorescence microscopy, and force spectroscopy. All injection experiments were conducted in low-fluorescence media. The initial experiments with the injection of LY were performed in PBS, at room temperature. To allow for viability assessment, the later experiments with the injection of AF488-H1 were performed in rich SD medium, at 30 °C for *S. cerevisiae*, and at 28 °C or 37 °C for *C. cinerea*.

**Quantification of the injected volumes**. Fluorescence images acquired through a 40× objective were analyzed in Fiji. A sample region of interest (ROI) and a non-overlapping background ROI were defined, and their area and mean fluorescence intensity were measured. The total fluorescence intensity was calculated by multiplying the sample area by the mean fluorescence intensity of the sample to which the mean fluorescence intensity of the background was subtracted. The obtained total fluorescence intensities were normed to an exposure time of 1 s.

A calibration curve to define the fluorescence intensity per volume was obtained by measuring the total fluorescence intensities of areas of different sizes of the FluidFM cantilever probe, with a known channel height of 800 nm. The fluorescent intensities were plotted against their respective volume, and the amount of fluorescence for a given volume was then obtained by linear regression.

For injected cells, the net fluorescence was obtained by subtracting the total fluorescence intensity before the injection from the one after injection.

**Extraction**. The reporter strain A43mutB43mut *cgl2p*-dTom, which expresses cytoplasmic dTomato under control of the *cgl2* gene promoter, was used. The probes were coated with Sigmacote (see above). Following spring constant calibration (k = 2.3 ± 0.5 N/m), the microchannel of the FluidFM probe was filled with mineral oil by application of overpressure in the SD medium. The probe was then immersed in the SD medium, and the probe sensitivity was calibrated (β = 66 ± 3 nm/V). A hyphal cell was selected under the optical microscope, and the probe was positioned next to the cell for conducting a force-controlled approach in contact-mode and retraction to 9 μm above the substrate. The tip of the probe was then moved above the hyphal cell and AFM force spectroscopy was initiated with a preset force of 2000 nN at 1.0 μm/s. Once inserted, the tip was maintained inside the cell by maintaining a constant force for 10–20 min. Maximum underpressure was applied to extract

cytoplasmic fluid into the probe. The pressure was then switched back to zero, and the probe was retracted out of the cell. Extraction experiments were conducted at room temperature (24 ± 1 °C).

**Quantification of *cgl2p*-dTomato expression**. Following the extractions ($N = 5$), the area of the mycelium that comprised the extracted hyphal compartments was monitored at room temperature (24 ± 1 °C) with a 10x objective and a 2x lens, in brightfield and with the 561 laser for the dTomato reporter. Z-stacks (200 slices, 300 nm) were acquired at 120 min intervals during 18 h. For each time-point, an image was created by summing the slices of the acquired dTomato Z-stacks in Fiji. dTomato fluorescence intensities were then measured as described above (Quantification of the injected volumes). Boundaries of individual hyphal compartments and backgrounds were all manually defined, and background intensities were measured for each time point and subtracted from all intensity values. We analyzed 5 extracted and 5 unperturbed hyphal compartments.

**Reporting summary**. Further information on research design is available in the Nature Research Reporting Summary linked to this article.

## Data availability
The datasets generated during and/or analysed during the current study are available from the corresponding author on reasonable request. Source data for the graphs and charts presented in this manuscript are available in the Supplementary Data.

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

## Acknowledgements

This work was supported by a grant from the Volkswagen foundation (Initiative "Life") to J.A.V., a European Research Council Advanced Grant (no. 883077) to J.A.V., and the Swiss National Science Foundation (Grant No. 31003A_173097) to M.K.

## Author contributions

O.G., T.Z., M.K., and J.A.V. designed the study. O.G., V.M., and S.S. designed, performed and analyzed the experiments. C.G.G. designed and manufactured the FluidFM-probes. O.G. and J.A.V. wrote the manuscript with input from all authors.

## Competing interests

The authors declare no competing interests.
