## [Peer Review File · Communications Biology]

Reviewers' comments:

Reviewer #1 (Remarks to the Author):

In their paper, Guillaume-gentil et al. propose a novel method based on FluidFM technology for injection or extraction into and from the cytoplasm of 4 species of fungal cells. They obtained the proof-of-concept by observing the delivery of Lucifer Yellow dye under sufficient applied force allowing to penetrate the resistant fungal cell-wall, and demonstrated that the cells remain alive after injection by following their growth. They also demonstrated the possibility of extraction of the cytoplasmic fluid and present an example of application to the biologically-relevant system *Coprinopsis cinerea*/nematode.

The paper is simply and clearly written, the proof-of-concept is elegant and the last example concretely shows the potential application of the method. I believe this interdisciplinary study is of interest for the community of Communications biology readers and I therefore recommend it for publication after minor corrections as mentioned below:

-On figure 1, the authors present indentation AFM curves for the 4 fungal cells. However, for each species, the force-distance curves present some profile which differ. Can it be that under such high applied force, the AFM probe goes through the nucleus? How would that translate on the indentation curve and what would be the consequence on the dye delivery for instance, or viability of the cells? This should be discussed in the manuscript.

- L 142, the authors claimed that the validation of the cytoplasmic location of the tracer is confirmed by the fact that the vacuoles remained unlabeled. However, it is not clear on the images where are the unlabeled vacuoles, this should be more clearly indicated.

- L 280, for *S. cerevisiae* at *S. pombe*, the authors used microstructures to ensure proper immobilization of the cells prior to injection. Although the patterns are clearly visible on image 2b, they are not easily seen of Fig. 2a, please explain why.

- On Figure 2a, it seems that the injected yeast is in an air bubble? Or do the images show one yeast stained in green, but in this case why are the surrounding cells smaller? The description of the image should be completed to help the reader.

Reviewer #2 (Remarks to the Author):

The paper by Guillaume-Gentil et al. reports on the injection of active materials into fungal cell. This is a very interesting paper using the state-of-the-art AFM technology. I recommend publishing this paper after addressing the following points:

1. Introduction – the authors assume that the readers are experts in the area of force spectroscopy. The introduction should be rewritten to fit the general audience of the journal.

1. In lines 89-95 the authors discuss how fungal cell stiffness affects indentation and the optimizations required for successful injection, and they refer to typical force vs. distance curves in figure 1b. It would be better if they would also explain the force curve, and how cell stiffness and penetration is reflected in these force profiles. In this regard, it would be very helpful if the authors added a figure (even a schematic supplementary figure) which shows the entire cycle of the injection experiment, with labels explaining each section, how cell indentation and penetration are detected, at what point in the experimental cycle is injection/extraction performed, and if and how the applied pressure is represented in the force curve, when the tip exits the cell and how that is reflected, etc.

Something of that sort: SEE ATTACHMENT

2. In figure 2a left panel, the *S. cerevisiae* cell shows some fluorescent signal before injection of LY. This low signal is only visible for the injected cell. Is this some trace of LY from a previous injection attempt?

3. In line 135 the authors refer to figure 1b regarding yeast cell sizes, should be figure 1c.

Reviewer #3 (Remarks to the Author):

Guillaume-Gentil et al paper 'Injection into and extraction from single fungal cells' describes an elegant and effective method that employs a modified AFM tip for selectively sampling from and injecting into individual fungal cells of differing morphology. While not a wholly novel idea, this work extends from a similar approach employing AFM developed by the same group for mammalian cells. However, the extensive modifications they describe here overcomes many of the specific challenges of trying to work with single fungal cells such as the resistance of cell walls, smaller cell sizes, different morphologies and non-adherence to surfaces.

The article addresses the fungal-specific challenges systematically and uses clear examples to demonstrate the AFM tip pressures required to effectively pierce the cell wall but still result in viable cells, modification of surfaces to maintain a grip of the cell during treatment, injection into cells of fluorescent fluids and quantity and impacts of removing cytoplasm. With the injection of fluids using the FluidFM approach, it is noted there was a 100x range of injected fluid into cells over the replicated experiments for each of the 4 fungal species examined. This is a large range of volumes. The authors could comment on this variation and what impacts they predict or expect of this.

This method will no doubt be adopted by many who want to be able to manipulate fungi at the single cell level. The article describes a minimum set of experiments that demonstrates the utility and effectiveness of the technique but lacks that extra depth or compelling information that may have come from, for example, demonstrating the fungal cellular fluid removal can provide sufficient material for single cell metabolomics or by injecting some kind of recombinant DNA/RNA to alter the phenotype once expressed or demonstrating intracellular binding of DNA/RNA etc. It is strange that these kinds of experiments were not described in tandem with the technique.

The paper is a pleasure to read, succinct and very well written. The methods are clearly described and sufficiently detailed.

Line 163 is missing an "injected" perhaps?

Reviewer #1 (Remarks to the Author):

In their paper, Guillaume-gentil et al. propose a novel method based on FluidFM technology for injection or extraction into and from the cytoplasm of 4 species of fungal cells. They obtained the proof-of-concept by observing the delivery of Lucifer Yellow dye under sufficient applied force allowing to penetrate the resistant fungal cell-wall, and demonstrated that the cells remain alive after injection by following their growth. They also demonstrated the possibility of extraction of the cytoplasmic fluid and present an example of application to the biologically-relevant system *Coprinopsis cinerea*/nematode.

The paper is simply and clearly written, the proof-of-concept is elegant and the last example concretely shows the potential application of the method. I believe this interdisciplinary study is of interest for the community of Communications biology readers and I therefore recommend it for publication after minor corrections as mentioned below:

-On figure 1, the authors present indentation AFM curves for the 4 fungal cells. However, for each species, the force-distance curves present some profile which differ. Can it be that under such high applied force, the AFM probe goes through the nucleus? How would that translate on the indentation curve and what would be the consequence on the dye delivery for instance, or viability of the cells? This should be discussed in the manuscript.

The force-distance profiles shown in Figure 1b are each from a different cell. Although the profiles are similar for a given species, they are indeed slightly different from each other due to inherent cell-to-cell variations (e.g. different cell size, rigidity, etc...). We did not specifically target the nuclei for injection due to their small size. In previous experiments with mammalian cells, we have indeed observed, as the reviewer suggests, that tip insertion and injection in the nucleus require higher forces than for the cytoplasm, and that cell viability was not compromised upon nuclear injection (Guillaume-Gentil et al., Small, 2013). Because the small molecule Lucifer yellow can pass through the nuclear pore complex by passive diffusion, both nucleus and cytoplasm were labelled upon injection, regardless of the injection site. In contrast, larger molecules that cannot pass through nuclear pores, such as 75kDa-TRITC-Dextran, located exclusively in the subcellular compartment they were injected in.

The discussion has been amended for clarification (page 8, lines 240-241): "Unlike mammalian cells, direct injection into fungal nuclei was not attempted as they were too small for insertion of the FluidFM tip."

- L 142, the authors claimed that the validation of the cytoplasmic location of the tracer is confirmed by the fact that the vacuoles remained unlabeled. However, it is not clear on the images where are the unlabeled vacuoles, this should be more clearly indicated.

We have amended Figure 2, panels c and d, to indicate where the vacuoles are located.

- L 280, for *S. cerevisiae* at *S. pombe*, the authors used microstructures to ensure proper immobilization of the cells prior to injection. Although the patterns are clearly visible on image 2b, they are not easily seen of Fig. 2a, please explain why.

To test the optimal microstructures for yeast immobilization, we designed arrays featuring a variety of micropattern geometries and dimensions, and the variable spacing between each microstructure. For the microwell shown in Figure 2a (1 microwell containing 2 yeasts), the other microwells of the array are outside of the field of view.

We have now added two Supplementary Figures, 2 and 3, to provide more details on the microstructure arrays used in this study.

- On Figure 2a, it seems that the injected yeast is in an air bubble? Or do the images show one yeast stained in green, but in this case why are the surrounding cells smaller? The description of the image should be completed to help the reader.

Figure 2a shows 1 oval microwell with two yeast cells, where only one of the cells was injected. This example was chosen to illustrate the selective labeling of the injected yeast, but not for example, the nonspecific binding of the dye to the cell wall that could have occurred if the injection failed. In the fluorescence images, the contour of the microwell is visible due to autofluorescence. The two yeast cells in the microwell, and the other, non-immobilized yeasts around it all have similar sizes.

We have amended the figure caption for clarity: "a) Bright field and fluorescence images of the injection of lucifer yellow (LY) into *S. cerevisiae*, before, during, and after injection. The images show a single oval microwell (in the center) containing two yeast cells; the right cell in the microwell was injected and was labeled, while the adjacent cell to its left was not injected and remained unlabeled."

Reviewer #2 (Remarks to the Author):

The paper by Guillaume-Gentil et al. reports on the injection of active materials into fungal cell. This is a very interesting paper using the state-of-the-art AFM technology. I recommend publishing this paper after addressing the following points:

1. Introduction – the authors assume that the readers are experts in the area of force spectroscopy. The introduction should be rewritten to fit the general audience of the journal.

1. In lines 89-95 the authors discuss how fungal cell stiffness affects indentation and the optimizations required for successful injection, and they refer to typical force vs. distance curves in figure 1b. It would be better if they would also explain the force curve, and how cell stiffness and penetration is reflected in these force profiles. In this regard, it would be very helpful if the authors added a figure (even a schematic supplementary figure) which shows the entire cycle of the injection experiment, with labels explaining each section, how cell indentation and penetration are detected, at what point in the experimental cycle is injection/extraction performed, and if and how the applied pressure is represented in the force curve, when the tip exits the cell and how that is reflected, etc.

Something of that sort: SEE ATTACHMENT

We agree that an additional figure will help the reader who is not familiar with force spectroscopy. We have now added a Supplementary Figure 1, which provides a detailed explanation of the force spectroscopy aspects of the present method.

2. In figure 2a left panel, the *S. cerevisiae* cell shows some fluorescent signal before injection of LY. This low signal is only visible for the injected cell. Is this some trace of LY from a previous injection attempt?

The initial fluorescence is caused by the weak autofluorescence signal from the edge of the microwell. The injected yeast shows a strong fluorescence signal only during and after injection. (Please see also the above response to the last 2 comments of reviewer #1)

3. In line 135 the authors refer to figure 1b regarding yeast cell sizes, should be figure 1c.

We thank the reviewer for noticing this mistake, and have now corrected it.

Reviewer #3 (Remarks to the Author):

Guillaume-Gentil et al paper 'Injection into and extraction from single fungal cells' describes an elegant and effective method that employs a modified AFM tip for selectively sampling from and injecting into individual fungal cells of differing morphology. While not a wholly novel idea, this work extends from a similar approach employing AFM developed by the same group for mammalian cells. However, the extensive modifications they describe here overcomes many of the specific challenges of trying to work with single fungal cells such as the resistance of cell walls, smaller cell sizes, different morphologies and non-adherence to surfaces.

The article addresses the fungal-specific challenges systematically and uses clear examples to demonstrate the AFM tip pressures required to effectively pierce the cell wall but still result in viable cells, modification of surfaces to maintain a grip of the cell during treatment, injection into cells of fluorescent fluids and quantity and impacts of removing cytoplasm. With the injection of fluids using the FluidFM approach, it is noted there was a 100x range of injected fluid into cells over the replicated experiments for each of the 4 fungal species examined. This is a large range of volumes. The authors could comment on this variation and what impacts they predict or expect of this.

A similar variation in injected volumes from cell to cell has been previously reported for microinjection with glass capillaries into mammalian cells (Minaschek et al., *Exp. Cell Res.*, 1989; [https://doi.org/10.1016/0014-4827\(89\)90402-3](https://doi.org/10.1016/0014-4827(89)90402-3)), and for FluidFM injection into mammalian cells (Guillaume-Gentil et al., *Small*, 2013). It has been mainly attributed to differences in the cytoplasmic viscoelasticity of the different cells, and the eventual presence of organelles or granules that may impede the release from the tip. In addition, small variations in the aperture size and the pressure pulse applied in this study may lead to further variations in the quantified volumes.

As a consequence of the inherent cell-to-cell heterogeneity, the accurate determination of the injected volumes can only be made after injection. If the injected molecule does not exhibit fluorescence, a fluorescent marker can be co-injected for downstream quantification.

We amended the discussion to add a comment on this point (page 8, lines 236-240): "A relatively large cell-to-cell variation in the injected volumes was observed, similar to what has been reported for injections into mammalian cells.^{14, 31} Therefore, determination of the injected volumes can only be done post-injection by fluorescence measurements. If the molecule of interest does not have fluorescence, it can be injected together with a fluorescent marker."

This method will no doubt be adopted by many who want to be able to manipulate fungi at the single cell level. The article describes a minimum set of experiments that demonstrates the utility and effectiveness of the technique but lacks that extra depth or compelling information that may have come from, for example, demonstrating the fungal cellular fluid removal can provide sufficient material for single cell metabolomics or by injecting some kind of recombinant DNA/RNA to alter the phenotype once expressed or demonstrating intracellular binding of DNA/RNA etc. It is strange that these kinds of experiments were not described in tandem with the technique.

As the scope of this study was to introduce the method to inject and extract fungal cells, we selected only one application to illustrate the potential of extraction (mimicking nematode feeding on a filamentous fungi), and one for the injection (the active transport to the nucleus of a protein injected into the cytoplasm). However, we agree with the reviewer that the study opens many applications of interest for both injection and extraction in the future.

To our knowledge, metabolite profiling from input volumes as low as 100 fL has never been achieved so far, and will require dedicated analytical development to reach the required sensitivity and adjust to the ultra-low volumes and input material; similarly, transcriptome and proteome profiling from such ultra-low input material will also require dedicated bioanalytical development.

We have initiated the development of bioanalytical methods to analyze cellular biopsies collected with the FluidFM, and have now shown, for animal cells, the feasibility of metabolite profiling (Guillaume-Gentil et al., *Anal Chem* 2017), and of genome-wide transcriptome profiling (Chen, W., Guillaume-Gentil, O. et al., *Molecular recording using Live-seq*, bioRxiv 2021, doi: <https://doi.org/10.1101/2021.03.24.436752>). Further developments for –omics analyses of living cells using cytoplasmic extracts are in progress.

Regarding the applications for injection, in this report we show the feasibility for small molecules (Lucifer yellow) and proteins (labelled histone H1). The process is expected to work similarly for any small molecules (e.g., drugs, metabolites), nucleic acids, or other proteins.

The paper is a pleasure to read, succinct and very well written. The methods are clearly described and sufficiently detailed.

Line 163 is missing an “injected” perhaps?

We thank the reviewer for pointing out this mistake, and have now corrected the sentence.

REVIEWERS' COMMENTS:

Reviewer #1 (Remarks to the Author):

The authors have addressed all my comments, I recommend publication in Communications biology

Reviewer #2 (Remarks to the Author):

As stated by the three reviewers, this is an interesting paper that should be published in this journal. The authors addressed all comments made by the reviewers and therefore I recommend accepting it as is.